# Genetic Therapy Approaches for Ornithine Transcarbamylase Deficiency

**DOI:** 10.3390/biomedicines11082227

**Published:** 2023-08-08

**Authors:** Berna Seker Yilmaz, Paul Gissen

**Affiliations:** 1Genetics and Genomic Medicine Department, Great Ormond Street Institute of Child Health, University College London, London WC1N 1EH, UK; p.gissen@ucl.ac.uk; 2National Institute of Health Research Great Ormond Street Biomedical Research Centre, London WC1N 1EH, UK; 3Metabolic Medicine Department, Great Ormond Street Hospital for Children NHS Foundation Trust, London WC1N 3JH, UK

**Keywords:** ornithine transcarbamylase deficiency, gene therapy, adeno-associated virus, messenger RNA

## Abstract

Ornithine transcarbamylase deficiency (OTCD) is the most common urea cycle disorder with high unmet needs, as current dietary and medical treatments may not be sufficient to prevent hyperammonemic episodes, which can cause death or neurological sequelae. To date, liver transplantation is the only curative choice but is not widely available due to donor shortage, the need for life-long immunosuppression and technical challenges. A field of research that has shown a great deal of promise recently is gene therapy, and OTCD has been an essential candidate for different gene therapy modalities, including AAV gene addition, mRNA therapy and genome editing. This review will first summarise the main steps towards clinical translation, highlighting the benefits and challenges of each gene therapy approach, then focus on current clinical trials and finally outline future directions for the development of gene therapy for OTCD.

## 1. Introduction

X-linked Ornithine transcarbamylase (OTC) deficiency (OTCD) [MIM: 311250] is the most common urea cycle disorder (UCD), accounting for almost half of the reported patients [1]. The prevalence of OTCD was estimated between 1 in 14,000 to 1 in 80,000 live births [2,3].

Ornithine transcarbamylase (OTC; EC 2.1.3.3) is a mitochondrial enzyme with the highest expression in the liver. Much lower expression levels of OTC exist in the small intestine and kidneys [4]. Hepatic OTC is essential for converting neurotoxic ammonia into non-toxic urea. It catalyses the synthesis of citrulline from carbamoyl phosphate and ornithine in the urea cycle. The enzyme is encoded by the human *OTC* gene, which is located on the short arm of the X-chromosome (Xp21.1), extends around 73 kb and comprises ten exons and nine introns [5,6]. More than 500 mutations have been reported in the *OTC* gene.

Phenotypes may be highly variable depending on the mutation, genetic background, and environmental factors. Due to the X-linked inheritance, heterozygous females may be asymptomatic [7,8]. While the total absence of enzyme activity is associated with severe neonatal phenotype, residual enzyme activity may lead to milder and late-onset phenotypes [9]. A recent study revealed that 19%, 12% and 7% of the patients had a neonatal-onset phenotype in France, Turkey, and the UK, respectively. In contrast, the most frequent primary symptoms were vomiting, altered mental status and encephalopathy in all countries [10]. In addition to the neurological presentation, liver presentation with acute liver failure or chronic liver disease is also common in OTCD [11].

The standard of care consists of a protein-restricted diet, ammonia scavengers, and arginine/citrulline supplementation. However, this treatment does not prevent acute hyperammonaemic episodes provoked by catabolic stress and can cause severe neurological impairment, coma and even death [12]. Until now, liver transplantation (LT) has been the only curative option; however, due to technical problems, lack of donor organs and requirement for lifelong immunosuppression, it is not broadly accessible [13,14]. Hepatocyte transplantation has also been proposed as a therapeutic approach in recent years. However, this does not provide a long-term solution and is used as a “bridge to transplant” [15]. Thus, disease management remains a challenge.

Gene therapy is a curative approach that corrects genetic defects by transferring or editing genetic material. In the past decade, gene therapy has become a disease-changing treatment for many inborn metabolism errors. OTCD has always been an attractive disease candidate for gene therapy using both viral and non-viral vector strategies [16]. Advances in genome editing tools, such as CRISPR/Cas9, have brought a new opportunity for mutation-specific and universal gene therapies, while viral and non-viral vectors may provide successful DNA and RNA delivery to the target cells. This review will outline the rationale of gene therapy for OTCD and elaborate on the improvement achieved from earlier phases to progressing clinical trials by underlining the pros and cons of individual delivery methods.

## 2. Why Is OTCD an Appealing Candidate for Gene Therapy?

OTC deficiency is most common among the UCDs and accounts for about half of the UCD patients [1]. There is a demographic variability. Studies identified an incidence of 1 in 62,000 in Finland, 1 in 63,000 in the USA, 1 in 69,904 in Italy and 1 in 80,000 in Japan [3,17,18,19]. However, this may be underestimated as milder cases may be undetected due to the late onset of subtle symptoms and the lack of sensitive and specific newborn screening [20,21].

There is a phenotypic range of severity varying from the neonatal onset to asymptomatic patients depending on the residual enzyme activity. Complete deficiency of OTC enzyme is mostly detected in hemizygous males, presents with severe episodes of hyperammonemia in the neonatal period and causes neurodevelopmental disability and high mortality [22]. Although all OTCD patients have a risk of hyperammonemia that can be triggered by stress, the ones with partial OTC deficiency present later in life, symptoms maybe be less severe or subtle, and they generally may have a better outcome [22,23]. It has been shown in a mouse model of OTCD that by increasing the minimal residual hepatocyte enzyme activity to 10% of normal, the phenotype can be improved from severe to mild one. Thus, total restoration of the physiological enzyme activity might not be required to improve disease phenotype. This makes OTCD an appealing target for liver-directed gene therapy [24].

A protein-restricted diet, ammonia scavengers, and arginine/citrulline supplementation constitute the current standard of care, which does not stop decompensation episodes and a poor quality of life [12]. LT restores OTC activity and leads to better neurological outcomes, especially when performed in early childhood [25,26]. However, it is technically more challenging to perform in children in the first three months of life or under 5 kg body weight complications may be more frequent, and survival rates may be lower in this group of patients [27,28]. LT cannot reverse the preceding neurological impairment. It must be done when patients are metabolically stable and, typically, without severe central nervous system (CNS) damage [12,27,29]. Thus, the ideal time for LT in classical neonatal cases of OTCD is only between 3 and 12 months of age. Moreover, LT brings its morbidity, mortality, and lifelong immunosuppression requirement, and there is a distinct shortage of donor organs [13].

Due to the chronic nature of the disease with frequent acute decompensation episodes and the lack of definitive treatment, OTCD causes a significant psychosocial and economic burden for patients, their caregivers and healthcare systems [30,31]. Overall, there is still a high unmet need for better therapies in OTCD.

Thus, OTCD is a feasible candidate for gene therapy development as it is a well-characterised monogenic disorder, and animal models are readily available for OTCD to allow preclinical testing.

## 3. Three Decades of Gene Therapy for OTCD

Gene therapy is based on transferring or editing a genetic material to treat a disease via nonviral or viral vectors. Gene therapy can be either in vivo by delivering a vector that packs the gene of interest or gene editing tools directly into a particular target tissue or blood circulation or ex vivo when patient cells are genetically modified outside the body and subsequently transplanted back into the patient. The most extensively used delivery system for in vivo gene therapy among viral vectors is adenoviruses (Ads) and adeno-associated viruses (AAVs) [16].

Ad is a non-enveloped virus with an icosahedral protein capsid and a 26- to 45-kb linear, double-stranded DNA genome that predominantly develops upper respiratory tract infections. The Ad genome is encircled by hairpin-like inverted terminal repeats (ITRs) that promote primase-independent DNA replication [32]. Up to now, over a hundred human Ad genotypes have been determined and classified into seven subgroups. Ad infection is initiated by interaction between the cell surface-localized receptors and the virus capsid, followed by endocytosis and endosomal escape. The viral DNA consequently goes into the nucleus, predominantly stays epichromosomal, and does not incorporate into the host cell genome [33].

Potential to transduce both replicating and quiescent cells, high transduction efficiency, large transgene capacity, non-integration into the host genome and broad range of tissue tropism are the significant advantages of Ads. However, high immunogenicity associated with a high prevalence of neutralizing antibodies (Nabs) from the pous infections, robust innate immunity to viral capsid, and strong adaptive immune reactions to de novo synthesized viral and transgene products are major drawbacks for Ads [34]. 

In 1995, a phase I pilot study (NCT00004386) was initiated, which used an Ad type 5 vector carrying human *OTC* cDNA administered into the right hepatic artery in adults with partial OTCD. Six different dose cohorts of three or four subjects received 2 × 10^9^ to 6 × 10^11^ particles/kg with 0.5 log increments. The first patient was treated on 7 April 1997, but the trial was terminated after the death of the 18th subject on 17 September 1999. Transgene expression in hepatocytes was shown in 7 of 17 subjects. Moderate elevation in ureagenesis capacity and/or fall in urinary orotate levels were detected in only 3 of the 11 subjects who had previous symptoms caused by hyperammonemia. Participants experienced several adverse events such as fevers, myalgias, nausea, emesis and biochemical abnormalities, including thrombocytopenia, anemia, hypophosphatemia, and elevated transaminases. All subjects developed neutralizing antibodies against the vector after administration, and antibodies persisted over a period of 18 months with a gradual decline [35].

The eighteenth participant was an 18-year-old male with partial OTCD diagnosed at 30 months. Despite having a protein-restricted diet and ammonia scavengers, he was experiencing multiple episodes of clinically symptomatic hyperammonemia. In the highest dose cohort, he received a dose of 6 × 10^11^ particles/kg. By 18 h after vector administration, he developed altered mental status and jaundice, which was not reported in the first 17 patients. After that, he demonstrated the signs of systemic inflammatory response syndrome and disseminated intravascular coagulation followed by multiple organ failure, leading to death at the 98th hour of gene transfer. A post-mortem examination confirmed numerous organ failures consistent with the cytokine storm triggered by the innate immune response against Ad capsid immediately after vector delivery [36].

A 66-year-old female patient with partial OTCD who received Ad gene therapy at 2 × 10^9^ particles/kg at 52 years on the same trial was diagnosed with hepatocellular carcinoma (HCC) [37]. Another subject from the same trial, a 45-year-old woman who received 2 × 10^10^ particles/kg at 45, was diagnosed with colon cancer 15 years later. No Ad vector sequences have been detected either in normal or tumor tissues obtained from these two patients, which suggests that Ad vector was not the direct cause of cancer in both patient [38].

After the early termination of the human clinical trial using early-generation Ads, helper-dependent adenoviral vectors (HDVs) have been developed to escape from the host cell-mediated immune response against viral proteins and prevent hepatotoxicity related to viral gene expression [39]. *hOTC* cDNA was inserted into a less immunogenic HDV under the control of the tissue-restricted promoter and combined with a post-translational overexpression strategy to correct the metabolic phenotype in adult OTC-deficient (*Spf^ash^*) mice. With a dose of 1 × 10^13^ particles/kg, a long-term metabolic correction was achieved with decreased orotic acid excretion, normal hepatic OTC activity, and raised OTC RNA and protein levels without chronic hepatotoxicity [40]. Nevertheless, because of the concerns about acute toxicity due to innate immunity, this gene therapy approach never progressed into human studies. Alternative approaches have been tested to decrease the minimal effective dose and increase the transgene expression. With the conventional infusion of an HDV containing a liver-specific enhancer (the hepatic locus control region from the apolipoprotein E gene (LCR)), a phenotypic correction was detected at a dose of 1 × 10^12^ particles/kg in *Spf^ash^* mice. By aiming for a further dose reduction, the same vector construct was delivered through a hydrodynamic injection technique and maintained phenotypic correction at a lower dose of 5 × 10^11^ particles/kg [41]. However, this type of hydrodynamic injection is unsuitable for humans, and no further translational work with Ads was undertaken in OTCD [42,43,44].

In the 2000s, AAVs became one of the most vigorously explored gene therapy vectors for safer in vivo delivery. AAV is a small (25-nm) virus from the Parvoviridae family with a non-enveloped icosahedral capsid that accommodates a linear single-stranded DNA genome of about 4.7 kb [45]. AAVs naturally infect humans but have not been connected with any known disease or illness until the discovery of a large number of children worldwide with severe hepatitis in the third year of the COVID-19 pandemic caused by simultaneous infection with wild-type AAV and adeno- or herpes viruses [46,47,48].

The AAV genome encodes four non-structural Rep proteins, three capsid proteins (VP1–3), an assembly-activating protein (AAP) and is sided by two AAV-specific palindromic inverted terminal repeats (ITRs; 145 bp) [45]. In the genome of recombinant AAV vectors, only the two ITRs (viral genome cis packaging signals) are maintained, while the remaining viral genes are changed with the exogenous DNA, which is known as a “transgene expression cassette” [49]. To date, 13 natural serotypes and over 100 variants of AAV have been identified [50]. Hybrid AAV vectors can also be generated by the transgene flanked by the AAV ITRs from serotype 2 (the first isolated serotype utilised as a gene therapy vector) and any accessible AAV capsids (Figure 1). In recent years, synthetically engineered AAV vectors have been generated to increase transduction efficiency, enhance vector tropism, avoid the host immune response and boost large-scale vector production [51].

The natural AAV serotype contains two open reading frames, Rep (replication) and Cap (capsid), flanked by two 145-base inverted terminal repeats (ITRs). In recombinant AAVs (rAAV), the viral genome is substituted with a synthetic expression cassette including a promoter, gene of interest and a terminator such as polyadenylation (polyA) sequence, flanked by the ITRs. To develop a pseudo-typed vector (e.g., AAV2/8), the cap genes from another AAV serotype (e.g., AAV8) can be utilised to pack the recombinant genome of another serotype (e.g., AAV2). Created with BioRender.com.

AAV viruses infect both replicating and non-replicating cells and remain in the host cell DNA by integration until a helper virus supplies the genes essential for its replication [45]. However, with recombinant AAV vectors, site-specific integration does not happen in the host DNA. It mostly remains episomal in the nucleus of transduced cells. Random AAV vector integration is reported in 0.1–1% of transduction events [52,53].

AAV vectors have several advantages as gene delivery tools. They have a vast extent of tissue tropism, are mostly non-integrative, can provide long-term gene expression and have a low immunogenicity profile in contrast to other viral vectors [54]. On the other hand, there are still various drawbacks that limit their clinical application. These include immunogenicity; potential for tumorigenicity; limited packaging capacity; and loss of transgene expression in a growing liver [53].

Although AAV vectors are weakly immunogenic compared to Adenoviruses, host immune response triggered by capsid or transgene is still a hurdle in clinical translation. AAV vectors can provoke both innate and adaptive immunity (Figure 2). Innate immune response activates the secretion of pro-inflammatory cytokines [55]. The complement system also plays a crucial role in innate defense, as several gene therapy trials reported adverse events associated with complement activation in the first few days after vector infusion [56,57]. B cells can produce Nabs against AAV capsid. Seroprevalence to different AAV types varies geographically, ranging from 20% to 80% of the population with a significant rate of cross-reactivity [58]. These antibodies, even at low titres, can neutralise the transduction of the target cells, especially if the vector is systemically administered. The presence of Nab titers over a certain titre is often an exclusion criterion for recruitment into clinical trials, which limits the availability of patients. It has also been shown that Nabs may remain even 15 years after AAV delivery, preventing effective vector re-administration [59,60]. Cell-mediated immunity against AAV capsid is another obstacle to persistent transgene expression. Dose-dependent T cell-mediated immune response caused a transient and asymptomatic increase in liver enzymes related to loss of transgene expression in several hemophilia trials [61,62,63]. Immunosuppression with oral corticosteroids is usually sufficient to dampen this response. However, it was not as effective in a recent hemophilia B trial [62,63,64]. 

AAV vectors can trigger both innate and adaptive immune responses. Innate immune responses induce the production of pro-inflammatory cytokines. The complement system also plays a crucial role in the innate immune defense. B cells can produce Nabs against AAV capsid. Viral vectors can induce CD8^+^ T cell responses to their antigens and a transgene product. Created with BioRender.com.

In neonatal mice, the transduction of actively replicating hepatocytes causes progressive loss of transgene expression [53]. In a neonatal Crigler-Najjar (CN) mouse model, either delaying the vector administration or giving higher doses of vector led to increased persistence of viral genomes and transgene expression [65]. Moreover, in diseases like Hemophilia, CN and UCDs, normal enzyme activity in approximately 10% of hepatocytes should be sufficient to convert the phenotype from a severe form to a milder one despite hepatocyte proliferation. There are also several studies testing strategies to succeed in dealing with the loss of AAV vector genomes in replicating cells by enhancing transgene integration into the host genome [66,67]. 

Small packaging capacity is limiting the use of AAVs as a gene therapy vector, as they cannot successfully pack more than 5 kb of DNA. While this causes a serious hurdle in treating diseases with mutations in significant genes, several approaches are in progress, like dual/triple vector and mini-gene strategies to achieve oversized gene transfer with AAVs.

There is a concern about AAV integration arising from the studies in mice and larger animals, indicating that AAV has a comparatively small risk of tumorigenesis [68,69]. Whilst a current study showed the integration of wild-type AAV2 genome fragments into the regions near the known proto-oncogenes in human hepatocellular carcinoma (HCC) specimens, these parts of the AAV genome are not present in the AAV vectors used for gene therapy [70]. An elevated incidence of HCC has been detected in the mucopolysaccharidoses type VII mouse model after AAV gene transfer in the perinatal period [71]. HCC was recorded as a participant in the HOPE-B gene therapy trial for hemophilia B in December 2020. The patient had no symptoms, and HCC was found on routine follow-up imaging. Whole genome sequencing of the tumor and bordering tissue revealed that AAV vector integration was probably not the reason for the tumor in this case. The patient had multiple risk factors for HCC, including a history of hepatitis B and C, nonalcoholic fatty liver disease, smoking, family history of cancer, and advanced age [72].

Several AAV-mediated studies have been done in OTCD mouse models. Adult OTC-deficient mice were treated with vectors pseudo-serotyped with the type 7, 8, and 9 capsids, and the AAV8 capsid was the most effective [73]. After that, a single intraperitoneal (i.p.) injection of an AAV2/8 vector encoding mouse OTC protein (mOTC) under the control of a liver-specific promoter resulted in a life-long and robust metabolic correction in adult male OTC-deficient *Spf^ash^* mice, while there was only a transient correction in the neonatal mice even with a double dose of the vector [74]. This was due to the loss of transgene expression in the growing liver.

As OTC-deficient *Spf^ash^* mice exhibit a mild phenotype without severe hyperammonemia episodes, the main therapeutic outcome is the correction of orotic aciduria. Therefore, a hyperammonemic mouse model of OTC deficiency was developed by short hairpin RNA (shRNA)-mediated knockdown of residual endogenous OTC mRNA. It has been proven that a fivefold lower dose of AAV2/8 construct was sufficient to prevent hyperammonemia compared to the one needed for orotic aciduria correction [75]. In a subsequent study using newborn i.p. injection of AAV2/8 construct, although there was a stable long-term expression in 8% of hepatocytes with a liver OTC activity around 34% of wild-type levels, it was found to be insufficient to control hyperammonaemia [24]. This highlighted the need for vectors that can provide long-term and stable gene expression. IV administration of an AAV2/8-based self-complementary (sc) vector expressing the mOTC gene under a liver-specific thyroxine-binding globulin (TBG) promoter increased OTC enzyme activity and vector genome copy numbers in the liver approximately 3-fold more compared to the single-stranded (ss) vector [76]. It was also reported that the difference in vector expression levels between sc and ss vectors was dose-dependent, and codon optimization was shown to be more successful in raising expression levels than using sc vectors [77]. Most recently, an AAV2/8 vector expressing a translationally codon-optimized human OTC cDNA (hOTCco) with the α1-AT liver-specific promoter enabled expression of human OTC and maintained long-term metabolic correction in a dose-dependent manner [74]. In further preclinical studies, a sc AAV8 vector that encodes hOTCco driven by a liver-specific promoter not only achieved metabolic correction but also prevented chronic liver damage and fibrosis [78]. It was also shown that multiple rounds of codon optimisation could be more beneficial compared to standard optimisation in terms of mRNA translatability and long-term therapeutic efficacy [79].

To overcome transgene loss and dilution in the growing liver, there was a focus on developing engineered AAV capsids with higher liver tropism [79,80]. AAVLK03 is an engineered capsid with 97.7% homology of the cap sequence and 98.9% homology of the amino acid sequence with the wild-type hepatotropic AAV serotype AAV3B [79]. AAVLK03 was shown to transduce human hepatocytes tenfold higher than AAV8 in chimera mouse-human livers and is more resistant to Nabs than wild-type capsids [79]. Moreover, the prevalence of antiAAVLK03 Nabs was relatively low in the pediatric population [58]. Intravenous gene transfer of AAVLK03 vector encoding the human codon-optimized OTC gene (AAVLK03.hOTC) was found to be safe in juvenile cynomolgus monkeys with an excellent liver tropism and sustained increase in the OTC enzyme activity. Only limited and transient humoral immune response was reported without cellular immune response [81].

Messenger RNA (mRNA) is also an emerging therapeutic tool for liver-based monogenic disorders. It can be translated into any type of protein, whether transmembrane, intramitochondrial, intracellular or secreted. mRNA can be encapsulated into lipid nanoparticles (LNP) to prevent rapid degradation by RNAses [80]. Delivery of mRNA has multiple potential advantages compared to viral vectors and other nucleic acid-based therapies. mRNA only needs to reach the cytoplasm to allow almost immediate initiation of protein translation. As it does not transit to the nucleus, there is no risk of genomic integration. mRNA LNP delivery can avoid the side effects frequently associated with viral delivery. Moreover, mRNA provides transient, half-life-dependent protein expression, which allows control of pharmacokinetics and dosing [80].

However, mRNA delivery still has some challenges to overcome and successfully transfer from bench to bedside. Firstly, large molecular size (10^5^–10^6^ Da) and being negatively charged decreases the penetration of mRNA over cellular membranes. Secondly, the half-life of mRNA is approximately 7 h, and it is an unstable molecule susceptible to degradation by RNAses. Therefore, the relatively short duration of protein production brings the need for repeated administrations. In addition, mRNA can also trigger an immune response, limiting the availability of mRNA-based therapeutics [80,81].

A recent study evaluated the efficacy and tolerability of Hybrid mRNA Technology delivery (HMT) for human OTC (hOTC) mRNA to OTC-deficient *Spf^ash^* mice. This a novel two-nanoparticle mRNA delivery approach including an N-acetylgalactosamine (GalNAc)-targeted polymer micelle for hepatocyte-specific delivery and endosomal escape and an LNP that keeps the mRNA safe from nucleases during delivery and entrance into the liver. Single IV injection led to an extensive mitochondrial hOTC protein expression all over the liver and elevated OTC enzymatic activity up to 10 days post-administration. Biweekly injections of hOTC mRNA resulted in normal plasma ammonia and urinary orotic acid levels and provided longer survival without toxicity [82].

## 4. Current Status of Clinical Gene Therapy Trials for OTCD

Over the past decade, in vivo, AAV gene addition and mRNA therapy have reached the clinical translation phase (Table 1). A phase 1/2, open-label safety and dose-finding study of AAV8-mediated gene transfer of human OTC in adult late-onset OTCD patients has completed recruitment (CAPtivate, NCT02991144). Three patients have received IV infusion of scAAV8OTC in each of three dose cohorts of 2.0 × 10^12^ GC/kg (Cohort 1), 6.0 × 10^12^ GC/kg (Cohort 2), and 1.0 × 10^13^ GC/kg (Cohort 3) and a fourth cohort, which included two patients receiving 1 × 10^13^ GC/kg with prophylactic steroids. Four of the five patients from the highest dose cohort are defined as responders and have remained clinically and metabolically stable. Overall, 7 out of 11 recruited patients stayed in a stable clinical and metabolic state. Six patients from the first three dose cohorts maintained stability in the 2–4.5 years follow-up period. Four complete responders have withdrawn ammonia-scavenger medications and a protein-restricted diet within one year [83]. In the cohort who received prophylactic steroids, one of two patients was determined to have a complete response. The second patient was recorded as a responder at week 36, but the status was changed to a non-responder from the 52-week visit. No treatment-associated adverse severe events, infusion-related reactions or dose-restricting toxicities were reported across all dose cohorts [84]. A long-term follow-up study planned for six years is ongoing (NCT03636438). The Phase 3 study is underway to explore the effect of scAAV8OTC for a 64-week primary efficacy analysis period. Fifty adolescent and adult patients will receive high-dose gene therapy or a placebo (NCT05345171) [85].

As paediatric patients represent the most at-risk population with severe long-term outcomes, they have significant unmet treatment needs. However, the clinical translation of AAV gene therapy for pediatric patients is still an ongoing concern. Higher efficacy and safety thresholds should be set to involve children as there may be age-related changes in pharmacokinetics and pharmacodynamics. Growing the liver is also challenging in the pediatric population as it may cause dilution of episomal transgene and a decline in transgene expression. Based on the successful preclinical studies with AAVLK03.hOTC, a phase 1/2 open-label, multicentric clinical trial targeting paediatric patients has been designed for clinical development (HORACE, NCT05092685) and is now at the pre-recruiting stage [85].

Based on the successful proof of concept studies for mRNA, a phase I/II clinical trial using single escalating doses of LNP hOTC mRNA was announced in OTCD patients. However, the trial was withdrawn before patient recruitment (NCT03767270). In 2020, another phase 1 randomized, double-blinded, placebo-controlled, ascending dose study assessing the safety, tolerability, and pharmacokinetics of single doses of OTC mRNA formulated in an LNP was initiated in healthy adults and completed recruitment (NCT04416126). With the same investigational medicinal product, a phase 1b study in adult OTCD patients (NCT04442347) and a phase 2 study involving both adolescent and adult OTCD patients (NCT05526066) are in progress.

## 5. Future Directions of Gene Therapy for OTCD

In recent years, genome editing tools have entered the gene therapy landscape. These are engineered nucleases that can alter the genome at a precise loci, involving zinc finger nucleases (ZFNs), transcription activator-like effector nucleases (TALENs), homing endonucleases (meganucleases) and Clustered Regularly Interspaced Short Palindromic Repeats (CRISPR)/(CRISPR-associated system) Cas9 [16]. These nucleases generate targeted DNA double-strand breaks (DSBs), which are then repaired by two major pathways. These pathways can be used to mediate gene correction, addition and deletion or disruption: Non-Homologous End Joining (NHEJ) means direct ligation of two DNA termini via random base insertions and/or deletions while Homology Directed Repair (HDR) precisely modifies the DNA in the presence of a donor repair template [86]. CRISPR/Cas9 is a promising in vivo and ex vivo gene therapy approach. The main challenge of the CRISPR/Cas9 system is the risk of generating ‘off-target’ mutations in genomic regions carrying similar sequences to the target site, which may cause undesirable side effects [87]. Moreover, it has been reported that up to 79% of analysed human samples have pre-existing antibodies, and 46% have T-cell immunity against Cas9 orthologues, increasing the chance of immune rejection [88]. 

Genome editing strategies have recently been explored for OTCD. Several studies reported the use of in vivo editing. A dual AAV system, including Cas9 and the other containing a guide RNA and the donor DNA, was tested in the OTC-deficient *Spf^ash^* mice. The mutation was corrected via HDR in 10% (6.7–20.1%) *OTC* alleles in the neonatal mice. However, gene correction was significantly lower in the adult mice with a reduced protein tolerance and fatal hyperammonemia on a chow diet [89]. In a further study, to establish a mutation-independent CRISPR-Cas9–mediated gene targeting process, a dual AAV vector system that includes a single guide RNA (sgRNA) regulated by the U6 promoter to target intron 4 of the *mOTC* locus and a functional minigene expressing codon-optimized human OTC (hOTCco) regulated by a liver-specific thyroxine-binding globulin (TBG) promoter (TBG.hOTCco.pA) has been used. Targeted vector administration in mice resulted in 25% and 35% of OTC-expressing hepatocytes at 3 and 8 weeks, respectively; this was four- and three-fold higher than the ones achieved with untargeted vector [90]. In another study, in vivo editing of patient-derived primary human hepatocytes with a dual AAV approach, using a highly human tropic NP59 capsid rather than AAV8, revealed successful editing in 29% of OTC alleles at clinically applicable doses. No off-target editing events were detected at the predicted sites in the genome [91].

CRISPR-based ex vivo gene editing strategy has also been assessed in OTCD [92]. Hepatocytes obtained from an OTCD patient were corrected ex vivo via the deletion of a mutant intronic splicing site achieving editing efficiencies >60% and the restorage of the urea cycle in vitro. The corrected hepatocytes were transplanted into the liver of FRGN mice and expanded to high levels (>80%). Transplanted animals showed normal ammonia, increased removal of an ammonia challenge, elevated OTC enzyme activity, and decreased urinary orotic acid levels. There was no sign of off-target editing [92]. While these preclinical studies are encouraging, some challenges must be overcome to achieve clinical translation.

## 6. Conclusions

Gene therapy is rapidly emerging as a viable option for OTCD. Although there are several limitations, AAV vectors are attractive candidates for OTCD as they have high safety, a wide range of infectivity, low immunogenicity, and long-term gene expression. Based on successful proof-of-concept studies for in vivo AAV delivery, clinical translation is in progress for adult and paediatric patients with two different AAV vectors. Preclinical studies with mRNA are also promising, while novel technologies, including gene editing, are actively being investigated and can become a game changer in managing this severe disorder.

## Figures and Tables

**Figure 1 biomedicines-11-02227-f001:**
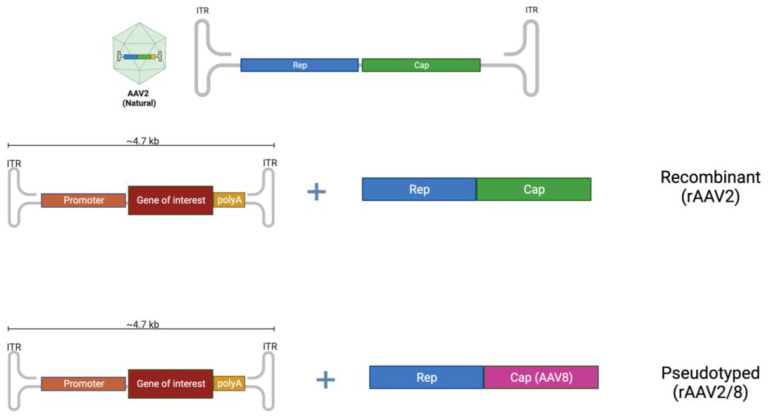
Adeno-associated virus (AAV) vector structure.

**Figure 2 biomedicines-11-02227-f002:**
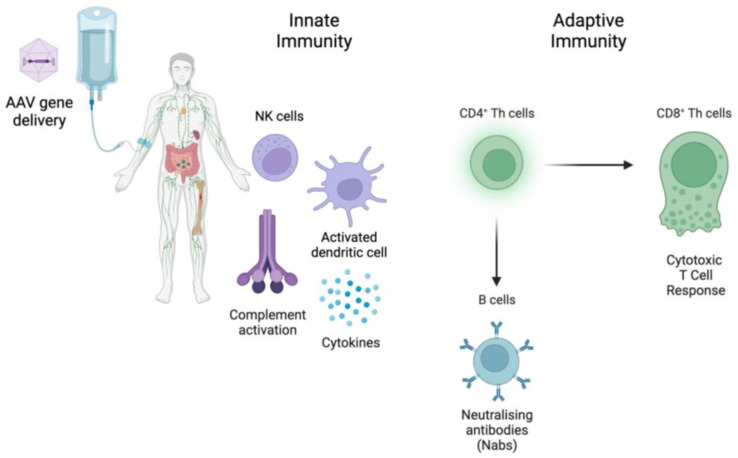
Immune responses against in vivo AAV delivery.

**Table 1 biomedicines-11-02227-t001:** Current gene therapy clinical trials for OTC deficiency.

Sponsor	Trial Number (clinicaltrials.gov)	Phase	Status	Vector	Number of Participants	Age Group of Participants (Years)
Ultragenyx (Novato, CA, USA)	NCT02991144	I, II	Completed	AAV8	16	≥18
Ultragenyx	NCT05345171	III	Recruiting	AAV8	50	≥12
University College London (London, UK)	NCT05092685	I, II	Not yet recruiting	AAVLK03	12	0–16
Arcturus Therapeutics, Inc. (San Diego, CA, USA)	NCT04416126	Ia	Completed	mRNA	30	18–65
Arcturus Therapeutics, Inc.	NCT04442347	Ib	Active, not recruiting	mRNA	12	≥18
Arcturus Therapeutics, Inc.	NCT05526066	III	Recruiting	mRNA	24	12–65

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
