# Peer review of "Genetic Therapy Approaches for Ornithine Transcarbamylase Deficiency"

_biomedicines, 2023, doi:10.3390/biomedicines11082227_

Round 1

Reviewer 1 Report

The review article is an updated overview of ornithine transcarbamylase deficiency and the therapies involved. It is well-written, to the point and with essential information for readers. 

Minor details refer to possible limitations of genetic therapy especially in this complex disease and in pediatric individuals.

Only minor details in the text were found

Author Response

06/08/2023

Dear Editor,

On behalf of all co-authors, I submit a revised version of the manuscript entitled “Genetic therapy approaches for Ornithine Transcarbamylase Deficiency” for your consideration for publication as a review article in Biomedicines special issue Hereditary Metabolic Diseases: The Biological Clock and Innovative Therapies

We thank the editors and reviewers for their time in assessing this manuscript. We found the reviewers’ comments helpful and we have addressed them extensively and in details. We hope our manuscript will now be judged suitable for publication.

Yours sincerely,

On behalf of all the co-authors,

Berna Seker Yilmaz

Response to Reviewer Comments

Reviewers' comments:

Reviewer 1:

The review article is an updated overview of ornithine transcarbamylase deficiency and the therapies involved. It is well-written, to the point and with essential information for readers.

Minor details refer to possible limitations of genetic therapy especially in this complex disease and in pediatric individuals.

Thank you very much for your positive feedback and remarkable suggestions. Limitations for the pediatric population have been further discussed as below.

‘As paediatric patients represent the most at-risk population with severe long-term outcome, they have significant unmet need for treatment. However, clinical translation of gene therapy for pediatric patients is still an ongoing concern. Higher efficacy and safety thresholds should be set to involve children as there may be age related changes in pharmacokinetics and pharmacodynamics. Growing liver is also challenging in pe-diatric population as it may cause dilution of episomal transgene and decline in transgene expression.’

Reviewer 2 :

The manuscript by Yilmaz and Gissen highlights genetic therapy approaches for ornithine transcarbamylase deficiency treatment. Due to the fact that this topic is still uncovered by any review, this manuscript is timely. The manuscript is well written and easy to follow. As a suggestion I would like to point authors attention on the fact that genetic therapy approaches are not restricted with AAV, mRNA or CRISPR/Cas9. Keeping this in mind, it seems that discussion of the issue of the optimal vector for gene delivery for OTCD treatment is missed. What are the benefits of AAV-based vectors in comparison to other vectors from the point of view of their use for OTCD treatment? What restrictions AAV has and what properties the ideal OTCD genetic therapy approach of the future should have?

Thank you very much for your valuable feedback and important suggestions.

Further discussion points have been added as below.

‘Although there are several limitations, AAV vectors are attractive candidates for OTCD as they have the advantage of high safety, wide range of infectivity, low immunogenicity and long-term gene expression.’

‘As paediatric patients represent the most at-risk population with severe long-term outcome, they have significant unmet need for treatment. However, clinical translation of AAV gene therapy for pediatric patients is still an ongoing concern. Higher efficacy and safety thresholds should be set to involve children as there may be age related changes in pharmacokinetics and pharmacodynamics. Growing liver is also challeng-ing in pediatric population as it may cause dilution of episomal transgene and decline in transgene expression.'

Reviewer 2 Report

The manuscript by Yilmaz and Gissen highlights genetic therapy approaches for ornithine transcarbamylase deficiency treatment. Due to the fact that this topic is still uncovered by any review, this manuscript is timely. The manuscript is well written and easy to follow. As a suggestion I would like to point authors attention on the fact that genetic therapy approaches are not restricted with AAV, mRNA or CRISPR/Cas9. Keeping this in mind, it seems that discussion of the issue of the optimal vector for gene delivery for OTCD treatment is missed. What are the benefits of AAV-based vectors in comparison to other vectors from the point of view of their use for OTCD treatment? What restrictions AAV has and what properties the ideal OTCD genetic therapy approach of the future should have? 

Author Response

(The authors gave the same response as above.)
